# Barcoding Fails to Delimit Species in Mongolian Oedipodinae (Orthoptera, Acrididae)

**DOI:** 10.3390/insects15020128

**Published:** 2024-02-12

**Authors:** Lea-Sophie Kock, Elisabeth Körs, Martin Husemann, Lkhagvasuren Davaa, Lara-Sophie Dey

**Affiliations:** 1Leibniz Institute for the Analysis of Biodiversity Change, University of Hamburg, Martin-Luther-King-Platz 3, 20146 Hamburg, Germanymartin.husemann@smnk.de (M.H.); 2Staatliches Museum für Naturkunde Karlsruhe (SMNK), Erbprinzenstraße 13, 76133 Karlsruhe, Germany; 3Department of Biology, School of Arts and Sciences, National University of Mongolia, P.O. Box 46A-546, Ulaanbaatar 210646, Mongolia; 4Senckenberg German Entomological Institute, Eberswalder Straße 90, 15374 Müncheberg, Germany

**Keywords:** orthoptera, cytochrome oxidase subunit I (COI), diversity

## Abstract

**Simple Summary:**

Barcoding is an easy way to differentiate taxa on the basis of the mitochondrial COI fragment. Databases like BOLD and NCBI house thousands of barcodes across all living taxa. In this paper, we provide sequences of the barcode region for most Mongolian band-winged grasshoppers. Furthermore, we analyzed their phylogenetic relationships and used different approaches for species delimitation. Taxa could only be differentiated at the tribe level, whereas species within genera were largely admixed. Similarly, species delimitation tools failed to cluster the taxa on a species level. We discuss several reasons for the lack of resolution including incomplete lineage sorting, nuclear copies of mitochondrial DNA, hybridization, and failing taxonomy.

**Abstract:**

Mongolia, a country in central Asia, with its vast grassland areas represents a hotspot for Orthoptera diversity, especially for the Acrididae. For Mongolia, 128 Acrididae species have been documented so far, of which 41 belong to the subfamily Oedipodinae (band-winged grasshoppers). Yet, few studies concerning the distribution and diversity of Oedipodinae have been conducted in this country. Molecular genetic data is almost completely absent, despite its value for species identification and discovery. Even, the simplest and most used data, DNA barcodes, so far have not been generated for the local fauna. Therefore, we generated the first DNA barcode data for Mongolian band-winged grasshoppers and investigated the resolution of this marker for species delimitation. We were able to assemble 105 DNA barcode (COI) sequences of 35 Oedipodinae species from Mongolia and adjacent countries. Based on this data, we reconstructed maximum likelihood and Bayesian inference phylogenies. We, furthermore, conducted automatic barcode gap discovery and used the Poisson tree process (PTP) for species delimitation. Some resolution was achieved at the tribe and genus level, but all delimitation methods failed to differentiate species by using the COI region. This lack of resolution may have multiple possible reasons, which likely differ between taxa: the lack of resolution in the Bryodemini may be partially explained by their massive genomes, implying the potential presence of large numbers of pseudogenes, while within the Sphingonotini incomplete lineage sorting and incorrect taxonomy are more likely explanations for the lack of signal. Further studies based on a larger number of gene fragments, including nuclear DNA, are needed to distinguish the species also at the molecular level.

## 1. Introduction

The large steppes of Mongolia are home to a wide variety of insects and other invertebrates. Grasshoppers make up a large proportion of these insects, especially in terms of biomass [1]. However, there are few studies dealing with the diversity of orthopterans in the country [2]. Among the 12,000 insect species known for Mongolia [3], 128 Acridid grasshoppers and 41 Oedipodinae species have been reported [2,4]. However, the knowledge on the insect fauna, and specifically the grasshopper fauna of Mongolia, remains limited.

The Oedipodinae have an almost cosmopolitan distribution. Up to now, around 800 species have been described [5,6]. Most of the species have colorful hind wing discs with or without black bands. They are an evolutionarily young group, which makes them particularly interesting for analyses of evolution, species delimitation, and hybridization, but they are also especially challenging to work with [7]. Further, they have some of the largest genomes in the insect world, rendering any kind of genetic studies in general more difficult [8,9].

To generate knowledge on the local flora and fauna of Mongolia, a cooperation between German scientists and the Committee of Science of Mongolia (now the Mongolian Academy of Science) was founded in 1961, which enabled a reference collection of local flora and fauna in Mongolia. Specimens found during the explorations were split between participating institutes in Mongolia and Germany [10]. In some of the later expeditions, Oedipodinae were investigated in particular [4,11], providing some more detailed data on the diversity and distribution of the group. Yet, so far no genetic data is available for most local species, not even DNA barcodes, which represent a minimum basis for genetic characterization of a species.

DNA barcoding is a fast and largely accurate method to identify species based on short, standardized gene regions, called barcoding loci [12]. The most commonly used barcoding locus in animals is the 648 bp long cytochrome c oxidase subunit I (Cox1; COI) [12]. In most animals, including many arthropods, COI sequences display sufficient differences to allow a cost efficient and easy way for species identification and differentiation [13]. Within the grasshopper community, the usage of barcoding data is torn. In several groups it is possible to differentiate the species [14,15,16], while in several other studies there is almost no resolution [17,18,19] which can have several reasons and will be discussed in this paper.

The Barcode Of Life Database [20], representing the global collection of DNA barcodes, holds 14,415 barcodes associated with 3344 Orthoptera species (last accessed October 2023, BOLD Systems) and therefore yields a good, yet incomplete (there are about 30,000 Orthopteran species globally), database for comparison and taxonomic evaluation. However, in order to really be useful in local projects, regional faunas need to be barcoded as completely as possible, and their data needs to be evaluated for their information content.

We here provide such a local study, where we generated barcodes of a large fraction of the local fauna of Oedipodinae and tested the data for its resolution at different taxonomic levels. Specifically, we performed the first local barcoding study for the Oedipodinae species of Mongolia. We generated COI sequences from specimens collected during several previous field trips to Mongolia and tested whether a difference could be determined. For this, we used multiple phylogenetic reconstruction and species delimitation methods.

## 2. Materials and Methods

### 2.1. Study Material

Sampling of Orthoptera for this study was done during field trips of LSD in 2015, 2017, and 2019 to Mongolia. All sampling was performed under government permit #200008 (2 January 2020). Specimens were identified morphologically following the identification keys by Mistshenko [21], Storozhenko et al. [22], and Dey et al. [23].

A total of 105 sequences of 35 species were obtained for this study, covering 83% of the known Oedipodinae species of Mongolia. The COI region of 82 individuals and 21 species was newly sequenced. A total of 10 species present in Mongolia according to Dey et al. [4] and Gankhuyag et al. [2] were not included in this study because of lacking material (Table 1). For *Sphingonotus coerulipes* and *Leptopternis gracilis*, sequences from individuals from Iran were used, due to lack of specimens of these species from Mongolia. Material of all Mongolian samples is stored at Senckenberg Germen Entomological Institute (SDEI), and the two samples from Iran are deposited in the State Museum of Natural History Karlsruhe (SMNK). All inventory numbers are given in Appendix A including species identification, sample location, and BOLD reference number.

In addition, 23 sequences were obtained from NCBI GenBank [24,25] and BOLD [20] (Table 2). One of these sequences (that of *Podismopsis altaica*, Gomphocerinae) was used as an outgroup in further analyses as it is known that Gomphocerinae are closely related to Oedipodinae. In further versions of this tree, more Gomphocerinae species were chosen as outgroup taxa; here, we only implement one taxon as outgroup, as the Gomphocerinae remained monophyletic in our tree. The sequences obtained from these online databases were those of species commonly found in Mongolia [2,4], but in some cases collected in surrounding countries.

### 2.2. Preparation of DNA for Sequencing

DNA was extracted from femur muscle tissue (stored in ethanol) using sterilized tweezers. After allowing the ethanol to evaporate, DNA was extracted via a Chelex extraction protocol (modified from Walsh et al. [26]). For each sample, 100 µL Chelex solution (5%) and 5 µL proteinase K (10 mg/mL) were added. All samples were then placed in a thermocycler with the following program: 1 h at 55 °C, 15 min at 99 °C, 1 min 37 °C. Extracts were stored at −20 °C.

The barcoding fragment of the mitochondrial COI gene was amplified using DreamTaq DNA polymerase (Thermo Fischer Scientific, Schwerte, Germany) in 10 µL final reaction volumes (5.8 µL ddH_2_O, 2.0 µL × 5 buffer w/Mg^2+^, 0.2 µL dNTPs, 0.4 µL COBU primer [27], 0.4 µL COBL primer [27], 0.2 µL *Taq* polymerase, 1.0 µL DNA). Primers were chosen based on previous studies performed on Orthoptera [14,28,29,30,31]. The PCR program included 30 cycles (3 min at 94 °C, 30× (30 s at 94 °C, 45 s at 48 °C, 1 min at 72 °C), 10 min at 72 °C). Afterwards PCR products were stored at −20 °C. The products were then run on a 1% agarose gel (25 min at 100 V) stained with Midori Green (Biozym, Hessisch Oldendorf, Germany) and visualized under UV light. Products which showed bands in electrophoresis were purified using the ExoCleanUp FAST Kit (VWR International GmbH, Darmstadt, Germany) and send to Macrogen Europe B.V. (Amsterdam, The Netherlands) for Sanger sequencing. All sequences were uploaded to BOLD Systems v. 3 [20] under project ID BMGL. All data entries and numbers are provided in Appendix A.

### 2.3. Sequence Analyses

Sequences were aligned using the MUSCLE (Multiple Sequence Comparison by Log- Expectation) [32] alignment tool in MEGA v.11 [33]. Frames were checked for internal stop codons. Furthermore all sequences were checked for contamination using BOLD Systems 4 [20] and blastN [34]

To choose the best-fitting substitution model, Modelfinder as implemented in IQTree web [35,36,37] was used. Furthermore, a maximum likelihood consensus tree was built in IQTree web using as sequence type Codon and implementing CODON5 (Invertebrate Mitochondrial) including 1000 replicates of ultrafast bootstrapping. The designated model was GTR + F + I + G4. The resulting tree was edited in FigTree v. 1.4.4 [38] and Inkscape [39]. In addition, we performed Bayesian inference analysis with MrBayes v. 3.2.7 [40]. We set the substitution model to GTR + F + I + G4. The MCMC was set to a chain length of 10,000,000, sampling every 1000 generations. Convergence was checked using Tracer v. 1.7.2 [41]. The final Bayesian consensus tree was then edited in FigTree v. 1.4.4 [38] and Inkscape [39].

Species delimitation was performed using two different methods: tree-based delimitation in PTP and distance-based delimitation with ABGD. Firstly, a Poisson tree process (PTP) model was run using the PTP webserver [42]. The program was run for 100,000 replicates, with a thinning of 100 and a burn-in rate of 0.1. Secondly, the dataset was analyzed using the ABGD webserver (Automatic Barcode Gap Discovery) [43] using the Jukes–Cantor model and a relative gap width of 1.5. Results of both species delimitation tools were visualized using the programs SPdel [44] and Inkscape [39].

## 3. Results

### 3.1. Maximum Likelihood

In the maximum likelihood tree (Figure 1), the outgroup *Podismopsis altaica* was used to root the tree. The tree recovered seven main groups representing the different tribes (Bryodemini, Sphingonotini, Locustini, Oedipodini, Epacromini, Parapleurini, and Gomphocerini) with low support.

The Bryodemini clade shows an admixed picture between the different genera. The support values are overall low (bs < 80) and the species of different genera are mixed up with no resolution even at the genus level.

The second large group comprises the species of the tribe Sphingonotini. While the support values are mostly high (bs 90–100), species remain mixed up across the group. Even the specimens of the genus *Helioscirtus*, and *Sphingoderus* are clustering within the specimens of the genus *Sphingonotus*. Furthermore there is a separated group of *Leptopternis* and both members of the species *Sphingonotus nebulosus* suggesting that the genus *Sphingonotus* is paraphyletic.

The tribe Locustini is monophyletic with high support values (bs 99–100) and with monophyly at the genus level. The genera *Oedaleus* and *Locusta* are separated. *Celes* as the only representative of the Oedipodini remains as a sister clade to the Bryodemini, Sphingonotini, and Locustini. The Epacromini represent the sister clade to all other tribes, including the monophyletic genera *Aiolopus* and *Epacromius*. This clade is supported by high bootstrap values of 99 to 100. The genus *Stethophyma* (Parapleurini) defines a sister clade to the Epacromini with a support value of 89.

### 3.2. Bayesian Inference Tree

The Bayesian inference tree generally shows less resolution (Figure 2). The species are separated at the tribe level. The Bryodemini clade is defined by lower posterior probabilities, and no clustering at the genus level is found. The Sphingonotini clade shows a better resolved grouping with most support values above 0.95, but still no separation at the species level. The species remain mixed up across the clade. The clades of Locustini, Oedipodini, Epacromini, and Parapleurini are supported by posterior probabilities above 0.95 but only include a low number of specimens. Based on this limited taxon set, the tribes remain monophyletic.

### 3.3. Species Delimitation

The PTP analysis suggests a separation into 35 mOTUs with high support for most of the groups (>0.9). Here, most species of the tribes Bryodemini and Sphingonotini were split into several mOTUs, which were admixed in their composition. The species of Epacromini, Locustini, Oedipodini, and Parapleurini were mostly correctly detected, except for *Celes variabilis* which was split into two species.

ABGD separated the dataset into 18 mOTUs, with two groups containing most of the Sphingonotini and Bryodemini individuals. *Leptopternis* and *Sphingoderus* are distinguished as separate units from the other Sphingonotini, while *Helioscirtus* is still part of a larger Sphingonotini unit. This analysis managed to differentiate the taxa of Epacromini, Locustini, Oedipodini, and Parapleurini as separate mOTUs. Similar to the PTP results, ABGD also placed the individuals of *Celes variabilis* into two groups. Figure 3 shows the results of the two species delimitation tools.

## 4. Discussion

In this study, we provide the first DNA barcode data for Mongolian Oedipodinae and tested the resolution of DNA barcoding for the group. Even though the species are morphologically readily distinguishable (except for some species within the genus *Sphingonotus*), the molecular results are not consistent with the morphological assignments. All phylogenetic reconstructions and species delimitation tools show the same overall results: specimens of the tribes Sphingonotini and Bryodemini cluster together as units, while the species and even genera are admixed. For the less diverse and not well represented tribes Epacromini, Oedipodini, Parapleurini, and Locustini, a separation is possible. In the following, we discuss our findings in more detail.

### 4.1. Basal Genera with Little Diversity

At the tribe level, barcoding provided a good resolution and the seven included tribes were monophyletic. Within the less diverse tribes of Mongolia, DNA barcoding provided an overall good resolution, e.g., *Celes variabilis*, *Stethophyma grossum*, *Locusta migratoria*, *Oedaleus infernalis*, *Aiolopus thalassinus*, and *Psophus stridulus* were recovered as monophyletic. This suggests that the species are obtained as monophyletic according to the barcoding data. In general, these genera include smaller numbers of species which are distributed across a large geographic range (the entire Eurasia and up to South Africa). Especially, in the target region, only single or few species are found and hence included in the analyses. As in other studies (e.g., Hawlitschek et al. [7], Moussi et al. [18]), such taxa with locally low diversity generally do not represent problems within barcoding studies of the group.

### 4.2. Bryodemini and Sphingonotini

While in some western and northern European countries, the local Orthopteran faunas are relatively poor and hence barcoding provides a sufficient resolution across large fractions of the species [7], this is the case when entering more diverse regions, such as the Mediterranean, Central Asia, or even the tropics. The admixed patterns we see in Bryodemini and Sphingonotini could have several reasons, for example, a young age of these tribes as suggested previously [45]. Interestingly, these two groups also include species with some of the largest genomes known so far [9], which may also point to some genomic scale effects potentially preventing lineage sorting or enhancing the rate of pseudogenes [46]. Furthermore as previously already described, the differentiation at the species and genus level has higher support values than the resolution of the backbone phylogeny [28,47,48]. Hence, in these groups, larger genome scale sampling of markers, or the development of lineage specific markers, may be required in the future to provide sufficient resolution at the higher levels. Based on only COI data, a clear statement is not possible, based on its frequent variability and its saturation

Besides these problems of marker resolution, the Sphingonotini in particular still include several species of unclear status. Many of the currently described central Asian species show very few identification traits and may be synonyms in some cases. Because these taxonomic issues are not solved, barcoding data will generally be difficult to interpret. Vice versa, based on the problematic use of barcoding data, it is rather complicated to check the definition of species due to monophyly of the clades. Hence, these nomenclatural problems will have to be solved until we get a better picture on the genetic diversity of these species groups.

### 4.3. Barcoding as a Solution for Species Delimitation?

Barcoding is a nowadays commonly used technique to determine species on the genetic level. Already in 2018, more than 3756 papers including the term barcoding in their title have been published [49]. Numbers are growing. But as useful it is maybe for some taxa, in other groups it is only of limited help. Even though in some grasshopper genera and geographical areas the method is promising to differentiate taxa, like in the Nearctic-Neotropical grasshopper genus *Taeniopoda* [15], in edible grasshoppers from Southern Africa [50], or in the Andean grasshopper genus *Orotettix* [51], it seems to be more complicated in several other grasshopper taxa e.g., in Acrididae [14,17].

The lack of resolution in these groups may have several non-exclusive reasons, i.e., incomplete lineage sorting, hybridization, NUMTs, imperfect taxonomy, and level of saturation in the COI region, and there is always a geographic bias [7,46,52]. Countries/regions with a high number of locally occurring species of a genus or tribe are more prone to an unresolved taxonomy based on barcoding data, while less diverse regions, like Germany, Austria, and Switzerland [7] do house a smaller number of species of one genus, often making it possible to differentiate them by using barcodes and species delimitation tools. There is also the possibility of frequent hybridization between species, as some of them are sympatrically distributed and share similar habitat types, which could lead to a replacement of mitochondrial genome between different species (mitochondrial capture) [53,54]. A further problem by using barcoding for species identification is the problem of imperfect databases. BOLD, for example, houses 60,159 barcodes of 3349 grasshopper species from around the world, of which 3733 belong to 225 species of Oedipodinae [BOLD systems, entered 5 January 2024]. Here, it is obvious that on one hand around 500 species are still missing, and on the other hand these specimens are not all correctly identified, which was already discussed by Lehmann et al. [55]. Hence, we think even though barcoding is not a solution for everything, it should be a major goal to feed the database with taxonomically correct data and simultaneously provide data of more gene fragments (mitochondrial and nuclear).

In our study, we show that the barcoding fragment is not adequate to differentiate the taxa at the species level, which may be the reason for its saturation level. These outcomes are akin to similar studies performed in other countries. Similar to our study, in Algeria, less diverse genera of Oedipodinae could be resolved by barcoding, while within the tribe Sphingonotini species remained admixed [18]. Similar patterns were reported in other studies of Orthoptera. For example, a study from China showed, based on a comparatively small dataset with species that are not very closely related, relatively good differentiation of species and tribes, but at higher levels they also had troubles to resolve the phylogeny [28]. A similar pattern was also shown for the Acridoidea from the Hebei Province in China [48]. Similar findings of badly resolved deeper nodes were also recovered in tetrigids. Here, a differentiation at species and genus levels were observed, while the COI fragment was too saturated to resolve the backbone phylogeny [47]. Again, other studies found much worse resolution at all levels: a study on the New Zealand endemic *Sigaus* showed some species splitting into several haplotypic clusters, while others shared one common haplotype. This admixture clearly shows how different grasshoppers are in their differentiation patterns of the COI region [17].

Summing up, the usage of only the barcoding fragment for species delimitation depends on the investigated organisms. For most grasshoppers, further gene fragments are needed to distinguish taxa at any level, as no barcoding study has been able to resolve deeper nodes with high support. Accordingly, further studies focusing on solving the phylogeny of Orthopterans are needed.

## Figures and Tables

**Figure 1 insects-15-00128-f001:**
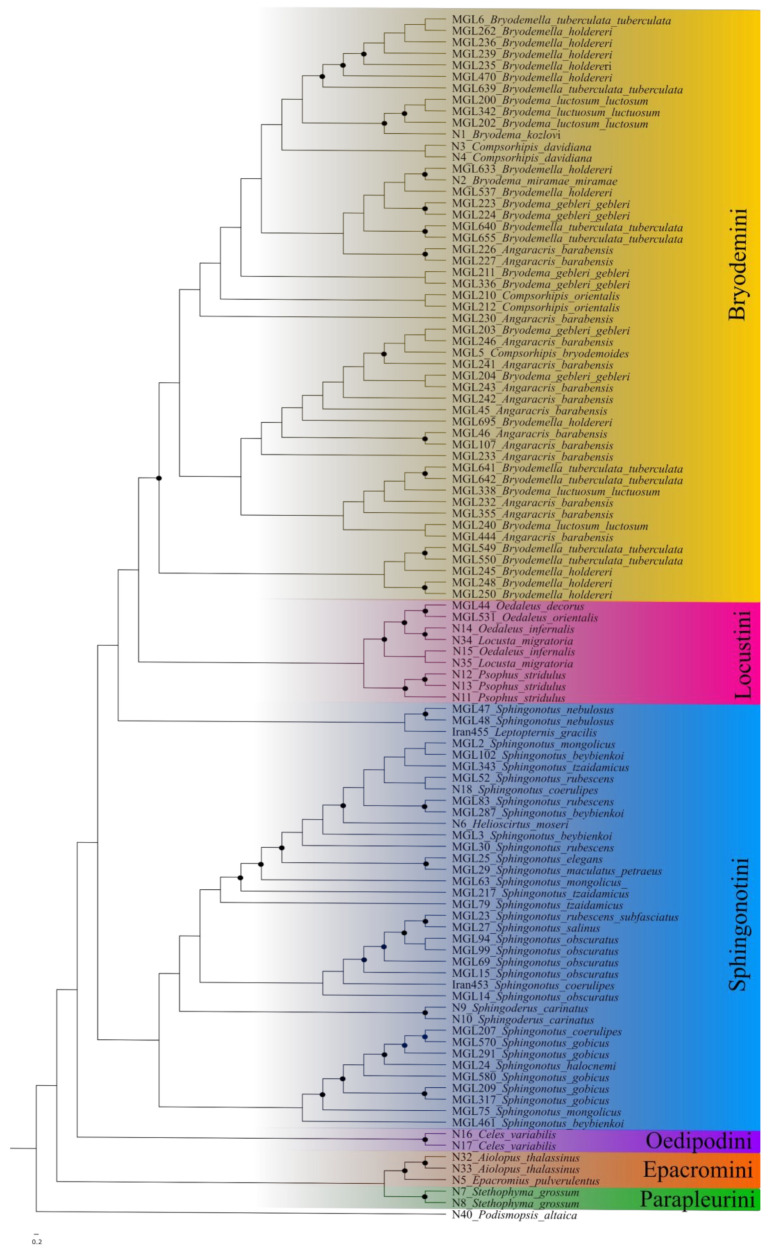
Maximum likelihood phylogeny based on the Oedipodinae dataset. Black dots represent nodes with bootstrap values above 90. Furthermore, results of PTP and ABGD species delimitation.

**Figure 2 insects-15-00128-f002:**
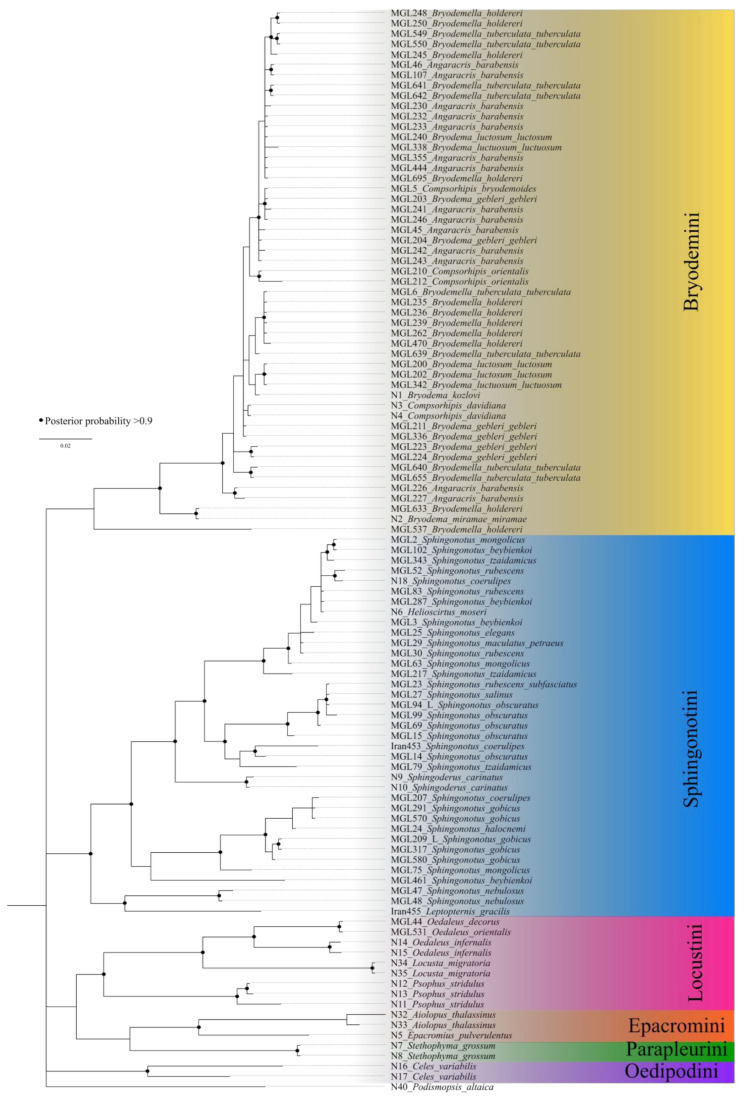
Bayesian Inference tree calculated with MrBayes. Black dots represent nodes with posterior probabilities above 0.9.

**Figure 3 insects-15-00128-f003:**
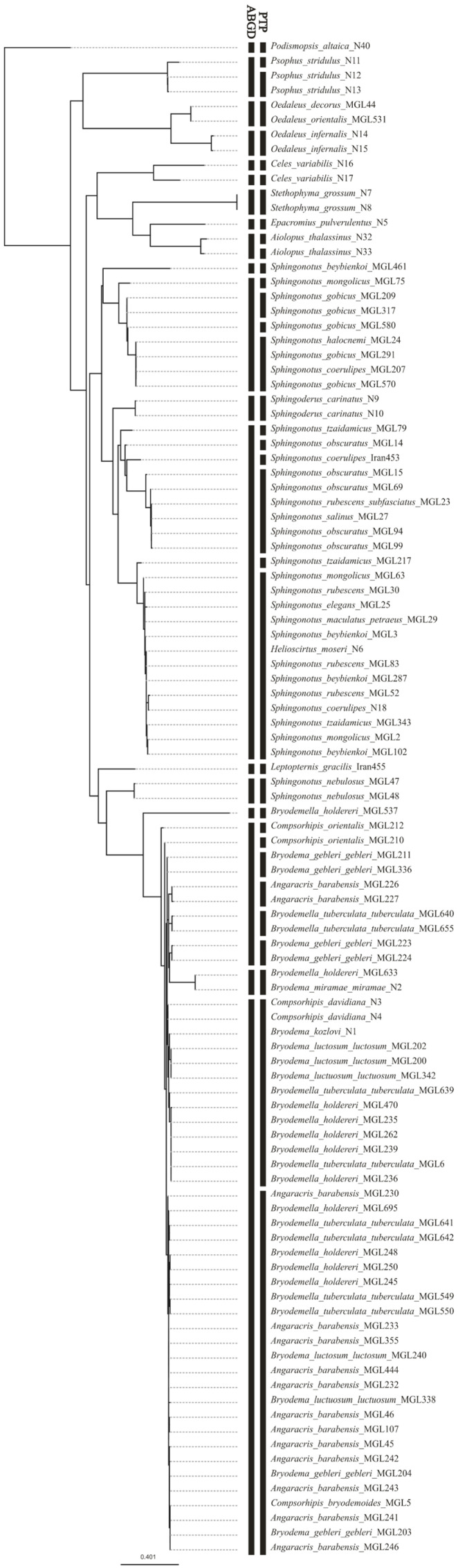
Maximum Likelihood tree compared with resulting clustering of the species delimitation tools (ABGD, PTP). Each black section defines one mOTU.

**Table 1 insects-15-00128-t001:** This table shows the missing species within this study. For these, no molecular data or specimens were available.

1	*Bryodema heptapotanicum*
2	*Bryodema nigripennis*
3	*Bryodemella orientale orientale*
4	*Bryodemella semenovi*
5	*Bryodemella zaisanicum fallax*
6	*Celes skalozubovi*
7	*Epacromius tergestinus tergestinus*
8	*Leptopternis iliensis*
9	*Sphingonotus lucidus*
10	*Sphingonotus halophilus*

**Table 2 insects-15-00128-t002:** COI sequences retrieved from NCBI GenBank and BOLD. ID refers to ID assigned in this study, species, origin of the sequence, and original ID from the database.

ID	Species	Origin	Original ID
N1	*Bryodema kozlovi*	NCBI	NC_052731
N2	*Bryodema miramae miramae*	NCBI	KP889242
N3	*Compsorhipis davidiana*	NCBI	NC_029408
N4	*Compsorhipis davidiana*	NCBI	KT157830
N5	*Epacromius pulverulentus*	NCBI	MT129326
N6	*Helioscirtus moseri*	NCBI	KR005923
N7	*Stethophyma grossum*	NCBI	GU706160
N8	*Stethophyma grossum*	NCBI	GU706136
N9	*Sphingoderus carinatus*	NCBI	MK251002
N10	*Sphingoderus carinatus*	NCBI	MK250997
N11	*Psophus stridulus*	NCBI	MT311126
N12	*Psophus stridulus*	NCBI	HQ955713
N13	*Psophus stridulus*	NCBI	GU706161
N14	*Oedaleus infernalis*	NCBI	KC297217
N15	*Oedaleus infernalis*	NCBI	NC_029327
N16	*Celes variabilis*	BOLD	GBMH5075_08
N17	*Celes variabilis*	BOLD	GBORT942_15
N18	*Sphingonotus coerulipes*	BOLD	GBMH5086_08
N32	*Aiolopus thalassinus*	NCBI	OQ214016.1
N33	*Aiolopus thalassinus*	NCBI	OQ214015.1
N34	*Locusta migratoria*	NCBI	OQ214122.1
N35	*Locusta migratoria*	NCBI	OQ214121.1
N40	*Podismopsis altaica*	NCBI	AY738343.1

## Data Availability

All barcoding data can be found on BOLD Systems v. 4.

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
