# Peer review of "Barcoding Fails to Delimit Species in Mongolian Oedipodinae (Orthoptera, Acrididae)"

_insects, 2024, doi:10.3390/insects15020128_

Round 1
Reviewer 1 Report
Comments and Suggestions for Authors
1. Some minor problems wih the English:
- page 1. In Abstract, “lineage” is misspelt as “linage”.
- Page 2. Last paragraph of Instruction. Instead of “In specific,”use “Specifically,”.
- In the next sentence I suggest following rephrasing: “ ...to Mongolia and tested whether a differentiation of the species is possible, based on the barcoding gene.”
- Page 3. Third paragraph. “From the obtained sequences one sequences of Podismopsis altaica (Gomphocerinae) served as outgroup for further analyses”. I suggest replacing with: “One of these sequences (that of Podismopsis altaica, Gomphocerinae) was used as an outgroup in further analyses”.
- Page 5. Figure 1. The tree is illegible in its upper regions (Bryodemini, some of Sphingonotini). I suggest adding an enlarged version (Fig. 1A) of these parts of the tree; you could get the necessary space for the branching pattern by abbreviating the generic names to the first initial letter (e.g., B. t. tuberculata, instead of Bryodema tuberculata tuberculata, et cetera). The Caption also needs attention: the sentence “Each black box represent a by the given method calculated OUT.” is unintelligible.
- Page 8. Section 3.3. Species delimitation. Last sentence. Replace “This analyses” with “This analysis”.
2. I think you should alter the title of this paper, to reflect its major finding: For example, it could read “Barcoding fails to delimit species in Mongolian Oedipodinae”. This would be more useful to the reader, especially if the reader were considering the use of bar coding.
Comments on the Quality of English LanguageThe standard of English is on the whole quite satisfactory. In "Comments to Authors" I have suggested a few improvements.
Author Response
Dear Reviewer,
thank you for your advice. We thoroughly revised our manuscript and integrated your mentioned suggestions. We also chose a differnt approach to make the ML Tree more vizuable.
Best
Lara Dey
Reviewer 2 Report
Comments and Suggestions for Authors
The paper by Kock et al on Oedipodinae species delineation using barcoding data is well-done, well-written and interesting for a large community, not only Orthoptera taxonomists.
I would advise however that the authors clearly distinguish between what the barcode data have been designed for (species delineation), and the interpretation that cannot be other than risky at higher taxonomic level. The genera recovered by barcoding data are monospecific, which does not mean that barcoding are valuable data to recover genera !
Barcode gene is short and easily saturated, but computer programs allow now to analyze these data as other phylogenetically significant molecular data. This does not mean that such analyses should be done at all taxonomic levels.
I would recommend publication with minor corrections dealing with : the deposit of sampled specimens (not indicated in the ms, unless I missed it), and more consideration about the adequation vs non-adequation of barcode sequences for phylogenetic conclusions. Studies at species and genus levels for example should be more cautiously compared.
Other editing errors and comments are indicated in the ms.

Author Response
Dear Reviewer,
many thanks for your suggestions. We integrated all the helpfull comments you made. Furthermore we also integrated a column with the species distributens in our appendix.
Best
Lara Dey
Reviewer 3 Report
Comments and Suggestions for Authors
I want to thank the authors for reporting their negative results. Not enough research teams do this.
Introduction
This study employs a conventional DNA barcoding approach using COI. The authors accomplish that goal and report their largely negative results. The paper is logically consistent overall. At the outset, however, I ask whether other loci were considered? I am suspicious of mitochondrial barcodes for Orthoptera because of the reasons the authors mention: huge genome sizes and rampant gene duplication. My own experience using COI for phylogenetic inference taught me that its utility is highly taxon specific in Orthoptera. The authors posit that barcoding is generally effective and this may be true across a diversity of animals, but in Orthoptera this claim is dubious. Orthoptera are a model system for numts. The North American grasshopper revision (that I am not involved in) abandoned mitochondrial markers. More can be said and cited about this, e.g. Song et al. 2014 discuss numts extensively and show just how pervasive the problem is.
ITS2 is an effective barcode for Orthptera, more so than COI in my experience, but some of the same problems occur with this locus as well e.g. Uluar & Ciplak 2020. Again locus utility is likely phylogenetically dependent and my bias is my work on Ensifera.
That being said, my suggestion here is that the authors expound upon the utility, or lack thereof, of COI barcoding in Caelifera in the Introduction and Discussion. The framing of this paper as a cautionary tale increases its utility to science. The authors often cite a European Orthoptera barcoding study, but the success of this study is perhaps more of an exception than a rule. Perhaps speculate upon and propose alternative barcodes that may improve results over COI.
Methods
Methods are generally well explained, but I would like to see a justification for the primer pair. First, PAN 2006 does not appear in the literature cited. Second, how were these primers chosen? There are numerous primer pairs for the barcode region, and these are not the conventional primers used for eDNA or published by Hebert in the original barcode literature. Primer choice may have a large impact on the results, and a good choice of primers or custom primers may go a long way toward avoiding numts and mitigating problematic results of the kind reported here. Folmer et al. primers are known to amplify numts, perhaps that is a reason the authors avoided them here? Have the authors aligned priming sites among taxa to investigate the building of custom primers? Please include any citations of exploratory primer usage, perhaps in previous work of the authors.
The authors mention screening products for internal stop codons, but numts may show no internal stop codons. There are several tools that identify pseudogenes/numts by screening PCR products against a consensus. I use ONTbarcoder to accomplish this for Oxford nanopore reads, and there are similar tools for Sanger sequences. I recommend the authors employ such a tool as an additional screening method.
Consider analyzing the data with a codon model. With a single locus, partitioning the data among codon positions may improve resolution. The GTR model is not likely to apply to more conserved codon positions.
Please provide a justification for the outgroup. Is this gomphocerine known to lie at a reasonable phylogenetic distance from the ingroup? Were multiple outgroups considered? Resolution may also depend upon outgroup choice.
Discussion
Band-wing grasshoppers are often behaviorally isolated, and when recognition signals fail hybridization may occur. The authors mentioned hybridization as a reason for the phylogenetic admixture they observed. Can the authors comment on the extent of geographic overlap or sympatry between some of the Mongolian band-wing taxa? A pattern may emerge where sympatric species are admixed, a common result resulting from mitochondrial capture between species boundaries.
The authors tend to cite previous works that they were part of and reference 7, the European Orthoptera barcoding paper, is cited often. Please expand citations to other research groups and other parts of the world, the barcoding literature is abundant, there are numerous options.
Comments on the Quality of English LanguageEnglish language is generally fine. Some minor corrections are:
line 95 spell out numbers that begin a sentence.
line 236 remove :"not"
Author Response
Dear Reviewer,
many thanks for your suggestions. We integrated all the helpfull comments you made. We also re-calculated our ML Tree with an integrated codon model. Furthermore we searched for tools which identify pseudogenes/numts, but there was no helpful tool. In the literature we mainly read that the people blast their sequences and look for internal stop codons. We added these information to the text.
We used Podismopsis as outgroup as Gomphocerinae are closely related to Oedipodinae. We also run the analyses with more outgroups and had the same results. Based on this we decided to keep it simple and only use one outgroup, as this shows enough resolution.
"Can the authors comment on the extent of geographic overlap or sympatry between some of the Mongolian band-wing taxa?" - yes we already mentioned the fauna study of mongolian Oedipodinae. In this publication we show the overlapping of the species in distribution maps.
Best
Lara Dey